# On the coherence of model-based dose-finding designs for drug combination trials

Yeonhee Park[1], Suyu Liu[2]*

**1** Department of Biostatistics and Medical Informatics, University of Wisconsin, Madison, WI, United States of America, **2** Department of Biostatistics, The University of Texas MD Anderson Cancer Center, Houston, TX, United States of America

* syliu@mdanserson.org

**Data Availability Statement:** All relevant data are within the manuscript and its Supporting information files.

**Funding:** The authors received no specific funding for this work.

## Abstract

The concept of coherence was proposed for single-agent phase I clinical trials to describe the property that a design never escalates the dose when the most recently treated patient has toxicity and never de-escalates the dose when the most recently treated patient has no toxicity. It provides a useful theoretical tool for investigating the properties of phase I trial designs. In this paper, we generalize the concept of coherence to drug combination trials, which are substantially different and more challenging than single-agent trials. For example, in the dose-combination matrix, each dose has up to 8 neighboring doses as candidates for dose escalation and de-escalation, and the toxicity orders of these doses are only partially known. We derive sufficient conditions for a model-based drug combination trial design to be coherent. Our results are more general and relaxed than the existing results and are applicable to both single-agent and drug combination trials. We illustrate the application of our theoretical results with a number of drug combination dose-finding designs in the literature.

## Introduction

The objective of phase I dose-finding clinical trials is to evaluate the safety and maximum tolerated dose (MTD) of new drugs through a sequential process of dose escalation and de-escalation. Various dose-finding designs have been proposed to guide dose escalation and de-escalation, but all are based on the same principle: if the observed data suggest that the current dose is safe (i.e., below the MTD), we escalate the dose to avoid treating the next cohort of patients at potentially sub-therapeutic doses; and if the observed data suggest that the current dose is overly toxic (i.e., above the MTD), we de-escalate the dose to avoid exposing patients to overly toxic doses. Because a phase I trial is the first-in-human trial and at that stage little known about the toxicity profile of the drug, it is critically important to ensure that the dose escalation and de-escalation rules are safe and appropriate. For single-agent trials, Cheung [1] introduced the concept of coherence, which is that a dose-finding design is coherent if it never escalates the dose when the most recently treated patient experiences toxicity, and never de-escalates the dose when the most recently treated patient does not experience toxicity. From

**Competing interests:** The authors have declared that no competing interests exist.

an ethical and practical viewpoint, it is desirable for a dose-finding design to be coherent. Cheung [1] established sufficient conditions for a single-agent dose-finding design to be consistent.

Drug combination trials have become a mainstream approach for treating cancer because of the ability to induce a synergistic treatment effect and overcome the drug resistance that is common with monotherapies. However, dose finding in drug combination trials is more challenging. In the dose-combination matrix, each drug combination has up to eight adjacent combinations as potential candidates for dose escalation and de-escalation. More important, these combinations are only partially ordered due to drug-drug interactions. For example, suppose one combination has a higher dose of drug 1 whereas the other combination has a higher dose of drug 2, *a priori* we do not know the toxicity order of these two combinations. As a result, when dose escalation or de-escalation is needed, it is not immediately clear which dose combination should be selected. In contrast, in single-agent trials, we are concerned with a string of doses that are associated with monotonically increasing toxicity. When dose escalation is needed, we simply move to the next higher dose level, and when dose de-escalation is needed, we move to the next lower dose level. Because of these fundamental differences between single-agent and multi-agent trials, the definition and results of coherence developed by Cheung [1] for single-agent trials cannot be directly applied to drug combination trials. In this article, we generalize the definition of coherence and establish sufficient conditions of coherence for drug combination trials. As we show, such generalization is not trivial and requires vastly different considerations. One important strength of our results is that they are still valid when the assumed drug-combination dose-toxicity model is misspecified.

Numerical dose-finding designs have been proposed for drug combination trials. The majority of them are model-based designs and make adaptive decisions of dose escalation/de-escalation using a strategy similar to that in the continuous reassessment method (CRM [2]). That strategy is to devise a parametric model to describe the dose-toxicity surface and then, based on the accumulating data, continuously update the model estimate to guide the dose selection and assignment. Thall et al. [3] proposed a six-parameter model-based design to find the MTD. Wang and Ivanova [4] developed a drug combination dose-finding design based on a log linear model. Yin and Yuan [5] and Yuan and Yin [6] proposed Bayesian dose-finding designs based on a copula-type regression model. Wages et al. [7] extended the CRM based on partial ordering of the dose combinations. Braun and Jia [8] proposed the generalized CRM model to guide dose finding. Riviere et al. [9] proposed a Bayesian dose-finding design based on the logistic model. Cai et al. [10] and Riviere et al. [11] adopted change-point models for drug combination trials involving molecularly targeted agents.

The remainder of the paper is organized as follows. In materials and methods section, we review coherence for single-agent trials and propose the generalized concept and theory of coherence that embraces both single-agent and drug combination trials. In application section, we use the proposed theory to study the coherence of a number of drug combination designs in the literature. In discussion section, we conclude with a discussion.

## Materials and methods

### Coherence for single-agent trials

We first review coherence for single-agent trials. Consider a single-agent phase I trial with $K$ doses under investigation, where a higher dose presumably has a higher probability of causing toxicity. Patients are sequentially enrolled, and each patient is treated at a dose that is adaptively selected by a certain trial design (e.g., CRM) based on the interim data. For each $n = 1$, ..., $N$, where $N$ is the prespecified maximum sample size, let $X_n$ denote the dose level assigned

to the $n$th patient, and $Y_n$ be a toxicity indicator of the $n$th patient, with $Y_n = 1$ denoting toxicity.

A design is called coherent in escalation if $\Pr(X_{n+1} > X_n | Y_n = 1) = 0$ and coherent in de-escalation if $\Pr(X_{n+1} < X_n | Y_n = 0) = 0$ for $n = 1, \ldots, N$ [1]. In other words, a coherent design never escalates the dose when the most recently treated patient experiences toxicity, and never de-escalates the dose when the most recently treated patient does not experience toxicity. Coherence provides a useful finite-sample–based metric to evaluate the safety and appropriateness of dose transition for phase I trial designs. Cheung [1] showed that the CRM is coherent when the prior estimate of the MTD is used as the starting dose.

## Coherence for drug combination trials

Consider a trial combining $J$ doses of agent 1, denoted as $u_1 < \cdots < u_J$, and $K$ doses of agent 2, denoted as $v_1 < \cdots < v_K$, where $u_j$ and $v_k$ are raw or standardized doses. Let $(j, k)$ denote the combination of $u_j$ and $v_k$, and $p_{j,k} = \Pr\{Y_n = 1 | X_n = (j, k)\}$ denote the probability of dose-limiting toxicity (DLT) for $(j, k)$. We assume that when the dose of one agent is fixed, the toxicity of the combination increases as the dose of the other agent increases, i.e., $p_{j,k} < p_{j',k}$ for $j < j'$ and $p_{j,k} < p_{j,k'}$ for $k < k'$. However, no toxicity order is assumed between other dose pairs. For example, the toxicity order between $(j, k)$ and $(j - 1, k + 1)$ and $(j + 1, k - 1)$ is unknown *a priori*. This partial order in toxicity makes drug combination trials fundamentally different from single-agent trials.

Consider a model-based dose-finding design that assumes a dose-toxicity model

$$F_{j,k}(\boldsymbol{\theta}) = F(u_j, v_k, \boldsymbol{\theta}), \tag{1}$$

where $F(\cdot)$ is a parametric model indexed by unknown parameters $\boldsymbol{\theta} = (\theta_1, \ldots, \theta_p)$. We assume that $\boldsymbol{\theta}$ is appropriately constrained such that the partial order is conformed, i.e., $F_{j,k}(\boldsymbol{\theta}) < F_{j',k}(\boldsymbol{\theta})$ for $j < j'$ and $F_{j,k}(\boldsymbol{\theta}) < F_{j,k'}(\boldsymbol{\theta})$ for $k < k'$. For example, for logistic model logit$\{F_{j,k}(\boldsymbol{\theta}) = \theta_1 + \theta_2 u_j + \theta_3 v_k + \theta_4 u_j v_k$, the partial order is imposed by the constraint $\theta_2 + \theta_4 v_k > 0$ for all $k$ and $\theta_3 + \theta_4 u_j > 0$ for all $j$. As the assumed dose-toxicity model may be misspecified, it is important to distinguish $F_{j,k}(\boldsymbol{\theta})$, the toxicity probability ascribed by the model, from the true toxicity probability $p_{j,k}$. Throughout the paper, we do not require $F(u_j, v_k, \boldsymbol{\theta})$ to be correctly specified.

Suppose that at an interim time, a total of $n$ patients have been enrolled and treated. Let $H_n = \{(X_i, Y_i), i = 1, \ldots, n\}$ denote the accumulative data from the enrolled patients. Unlike single-agent trials, where there is only a single candidate dose level $X_n + 1$ for escalation and $X_n - 1$ for de-escalation, in drug combination trials, there are multiple candidate doses are available for escalation or de-escalation. For example, in the case of dose escalation, we can escalate either the dose level of drug 1 or the dose level of drug 2. Therefore, it is imperative to generalize the definition of coherence.

Specifically, given $X_n$, let $\mathcal{E}_n$ and $\mathcal{D}_n$ denote the set of candidate dose levels for escalation and de-escalation, respectively, for the $(n + 1)$th patient, and define $\mathcal{A}_n = X_n \cup \mathcal{E}_n \cup \mathcal{D}_n$ to present all possible doses assignments for the $(n + 1)$th patient. A generalized definition of coherence that is applicable to both single-agent and drug combination trials follows.

**Definition 1** *A design is coherent in dose escalation if* $\Pr(X_{n+1} \in \mathcal{E}_n \mid Y_n = 1) = 0$ *for* $n = 1, \ldots, N$, *and is coherent in dose de-escalation if* $\Pr(X_{n+1} \in \mathcal{D}_n \mid Y_n = 0) = 0$ *for* $n = 1, \ldots, N$. *A design is coherent if its dose escalation and de-escalation are both coherent.*

The definition of coherence for the single-agent trial proposed by Cheung [1] is a special case of the above, with $\mathcal{E}_n = \{X_n + 1\}$ and $\mathcal{D}_n = \{X_n - 1\}$.

Given the observed interim data $H_n$ and a specific definition of $\mathcal{E}_n$ and $\mathcal{D}_n$, $X_{n+1}$ is typically determined as

$$X_{n+1} \equiv \underset{(j,k) \in \mathcal{A}_n}{\operatorname{argmin}} |F_{j,k}(\hat{\boldsymbol{\theta}}_n) - p_T|, \tag{2}$$

where $p_T$ is the target toxicity probability, $F_{j,k}(\hat{\boldsymbol{\theta}}_n) = F(u_j, v_k, \hat{\boldsymbol{\theta}}_n)$ is the model-based toxicity estimate, and $\hat{\boldsymbol{\theta}}_n = E(\boldsymbol{\theta}|H_n)$ is the posterior mean of $\boldsymbol{\theta}$. That is, the next patient will be treated at the dose level that belongs to $\mathcal{A}_n$ and has estimated toxicity probability closest to the target $p_T$. For the first patient, $H_n$ is empty and $F_{j,k}(\hat{\boldsymbol{\theta}}_n)$ is obtained based on the prior distribution of $\boldsymbol{\theta}$. In the case that it is desirable to start the trial from the lowest dose $(1, 1)$, we can set the prior estimate of the MTD as $(1, 1)$.

In drug combination trials, the choice of $\mathcal{E}_n$ and $\mathcal{D}_n$ has a profound impact on the coherence of a design, which does not arise in single-agent trials. To be consistent with practice, we herein assume the no-dose-skipping rule that restricts dose escalation and de-escalation to doses that are adjacent to the current dose. More precisely, given $X_n = (j, k)$, $\mathcal{E}_n \cup \mathcal{D}_n = \{(j+1, k), (j, k+1), (j-1, k), (j, k-1), (j-1, k+1), (j+1, k+1)\}$. We do not consider $(j+1, k+1)$ because dose escalation from $(j, k)$ to $(j+1, k+1)$ is equivalent to skipping doses $(j+1, k)$ and $(j, k+1)$, noting that $p_{j,k} < p_{j+1,k}$, $p_{j,k+1} < p_{j+1,k+1}$. For a similar reason, we do not consider dose de-escalation from $(j, k)$ to $(j-1, k-1)$. Nevertheless, all the results presented below are valid if $(j+1, k+1)$ and $(j-1, k-1)$ are included. In the literature, there are two common ways to specify $\mathcal{E}_n$ and $\mathcal{D}_n$, as illustrated in Fig 1. The first approach, referred to as the ND-design, does not allow for dose movement along the diagonals: $\mathcal{E}_n = \{(j+1, k), (j, k+1)\}$ and $\mathcal{D}_n = \{(j-1, k), (j, k-1)\}$. The second choice, the D-design, allows for dose movement along the diagonals, where $\mathcal{E}_n \cup \mathcal{D}_n$ contain doses along the diagonal, i.e., $(j-1, k+1)$ and $(j+1, k-1)$.

**ND-design.** We first investigate the coherence of the ND-design and refer to the following condition.

*Condition A.* For any $\boldsymbol{\theta} = (\theta_1, \ldots, \theta_p)$ and for any $n$ with $X_n = (j, k)$,

$$(Y_n - p_T)\left(\hat{\theta}_{n,t} - \hat{\theta}_{n-1,t}\right) \frac{\partial F(u_j, v_k, \boldsymbol{\theta})}{\partial \theta_t} \geq 0 \quad \text{for all } t = 1, \ldots, p, \tag{3}$$

where $\hat{\theta}_{n,t}$ denotes the $t$th element of $\hat{\boldsymbol{\theta}}_n$.

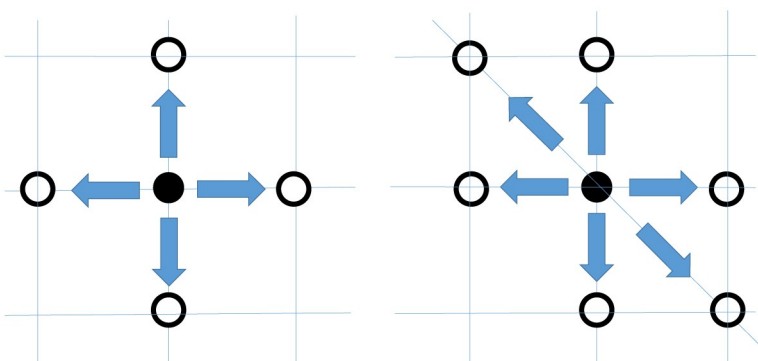

**Fig 1. (a) ND-design, which does not allow for diagonal dose movement, and (b) D-design, which allows for diagonal dose movement.** The filled circle indicates the current dose combination; the empty circles denote the candidate doses for escalation and de-escalation under the given design.

**Theorem 1** *Under condition A, the ND-design is coherent.*

Condition A requires the determination of the sign of $\hat{\theta}_{n,t} - \hat{\theta}_{n-1,t}$, for $t = 1, \ldots, p$, which may not be straightforward. A special case of condition A that is easier to evaluate is uniform monotonicity.

**Definition 2 (uniform monotonicity)** *$F(u, v, \boldsymbol{\theta})$ is uniformly nondecreasing (or nonincreasing) if for any value of $(u, v)$, $F(u, v, \boldsymbol{\theta})$ is nondecreasing (or nonincreasing) in $\theta_t$ for all $t = 1, \ldots, p$.*

**Lemma 2** *If F is uniformly monotonic, then condition A holds.*

**Theorem 3** *The ND-design is coherent if $F(u, v, \boldsymbol{\theta})$ is uniformly monotonic.*

The proof of Lemma 2 is provided in the Appendix A of Supplementary material. Theorem 3 is established directly from Lemma 2 and Theorem 1. Although uniform monotonicity is easier to check, many dose-toxicity models do not satisfy that condition, as shown later. In these cases, Theorem 1 can be used to examine the coherence of the ND-design.

**D-design.**  The D-design allows for dose escalation and de-escalation to the adjacent doses on the diagonals, i.e., $(j + 1, k − 1)$, $(j − 1, k + 1)$, assuming that $(j, k)$ is the current dose level. The complication is that *a priori* we do not know the toxicity order between $(j, k)$ and $(j + 1, k − 1)$ and $(j − 1, k + 1)$, thus it is not clear whether $(j + 1, k − 1)$ and $(j − 1, k + 1)$ belong to $\mathcal{E}_n$ or $\mathcal{D}_n$. Different definitions of $\mathcal{E}_n$ and $\mathcal{D}_n$ lead to different definitions of coherence. A straightforward approach is to define $\mathcal{E}_n$ and $\mathcal{D}_n$ based on the true toxicity probability of these doses. We refer to the resulting coherence as strong coherence, which indicates that the design will never escalate to a dose that is truly more toxic than the current dose if the most recently treated patient has toxicity; and will never de-escalate to a dose that is truly less toxic than the current dose if the most recently treated patient does not have toxicity.

To establish sufficient conditions for a design to be strongly coherent, we define conditions B1 and B2 as follows.

*Condition B1.* For any $\boldsymbol{\theta}$, $F(u, v, \boldsymbol{\theta})$ is increasing in both $u$ and $v$.

*Condition B2.* For any $j$ and $k$ and for any $\boldsymbol{\theta}$,

$$\text{sgn}\{F(u_{j+1}, v_{k-1}, \boldsymbol{\theta}) - F(u_j, v_k, \boldsymbol{\theta})\} = \text{sgn}(p_{j+1,k-1} - p_{j,k})$$

and

$$\text{sgn}\{F(u_{j-1}, v_{k+1}, \boldsymbol{\theta}) - F(u_j, v_k, \boldsymbol{\theta})\} = \text{sgn}(p_{j-1,k+1} - p_{j,k}),$$

where the signum function is defined by $\text{sgn}(x) = − 1, 0$ or $1$ for $x < 0$, $x = 0$ or $x > 0$, respectively.

**Theorem 4** *Under conditions B1 and B2,*

1. *If condition A holds, the D-design is strongly coherent.*

2. *If the condition of uniform monotonicity holds, the D-design is strongly coherent.*

The proof of Theorem 4 is provided in the Appendix A of Supplementary material.

As the true toxicity probability is unknown in practice, a more practical approach is to define $\mathcal{E}_n$ or $\mathcal{D}_n$ based on the model estimates. Recall that to determine dose level $X_{n+1}$ for the $(n + 1)$th patient, the model-based design fits the model $F(u_j, v_k, \boldsymbol{\theta})$ using interim data $H_n$, and then makes the decision of dose escalation or de-escalation for $X_{n+1}$ based on $F_{j,k}(\hat{\boldsymbol{\theta}}_n)$, as specified by Eq (2). Based on $F_{j,k}(\hat{\boldsymbol{\theta}}_n)$, the toxicity order of $(j + 1, k − 1)$ and $(j − 1, k + 1)$, with respect to $(j, k)$, can be determined and used to define $\mathcal{E}_n$ and $\mathcal{D}_n$. We refer to the resulting coherence as weak coherence. Accordingly, weak coherence indicates that the design will

never escalate to a dose for which the estimated toxicity probability is higher than the current dose if the most recently treated patient has toxicity; and will never de-escalate to a dose for which the estimated toxicity probability is lower than the current dose if the most recently treated patient does not have toxicity. Because of potential model misspecification, a weakly coherent design is not necessarily strongly coherent. For example, suppose $X_n = (j, k)$, $Y_n = 0$ and $X_{n+1} = (j + 1, k - 1)$. If $F_{j+1,k-1}(\hat{\boldsymbol{\theta}}_n) > F_{j,k}(\hat{\boldsymbol{\theta}}_n)$, but the truth is $p_{j+1,k-1} < p_{j,k}$, then the dose assignment for the $(n + 1)$th patient is weakly coherent, but not strongly coherent. This phenomenon is unique in drug combination trial designs because of the unknown toxicity order between some drug combinations. This phenomenon does not exist for single-agent trial designs, where the toxicity order is completely known among all doses. Despite this issue, weak coherence is still useful because the true toxicity probability is unknown in practice. From the user's perspective, it is certainly concerning if a trial design has a high likelihood of escalating to a dose that is expected to have higher toxicity after the most recently treated patient experienced toxicity at a lower dose. It can be shown that the D-design is weakly coherent under the same condition as the ND-design (see the Appendix A of Supplemental material for the proof).

**Theorem 5** *If condition A or the condition of uniform monotonicity holds, the D-design is weakly coherent.*

Comparing Theorem 5 with Theorem 4, it is clear that weak coherence is more lenient than strong coherence. With the extra requirements specified by conditions B1 and B2, a weakly coherent D-design becomes strongly coherent.

**A two-stage design.**   The above results are established under the assumption that the design starts by treating the first patient at the prior estimate of the MTD and the dose for each subsequent patient is selected based on the model estimates according to Eq (2). In practice, however, for patient safety, the trial often starts with the lowest dose, which is not necessarily the prior estimate of the MTD. In addition, as the dose-toxicity model for drug combination trials is relatively complicated, to improve estimation and design reliability, the two-stage design is often used. In the first stage (or the initial design), we make the decision of dose escalation/de-escalation based on a set of simple prespecified rules, without using the model, to collect some preliminary data. We then switch to the second stage (or the model-based design), in which we base the decision of dose escalation/de-escalation on the model estimates, as described previously. A typical example of the initial design is the "titration" design, under which we pre-select a string of dose combinations with monotonically increasing toxicity; see Fig 2. We treat the first patient at the lowest combination (1, 1) and then escalate the dose for treating the next patient if no toxicity is observed for the current patient. We continue this dose escalation (or titration) process until we encounter the first toxicity, and then we switch to the model-based design. Cheung [1] studied the condition of coherence for two-stage single-agent trials. In what follows, we provide the coherence condition for two-stage combination trials.

Let $X_{1n}$ and $X_{2n}$ denote the dose assignment according to the initial and model-based designs, respectively. Let $R$ denote a prespecified rule that triggers the switch from the initial design to the model-based design. Then, the dose assignment for the two-stage design, denoted as $X_n^*$, can be formally defined by

$$X_n^* = \begin{cases} X_{1n} & \text{if } R \text{ is not satisfied} \\ X_{2n} & \text{if } R \text{ is satisfied} . \end{cases}$$

The theorem below provides a sufficient condition for the two-stage design to be coherent.

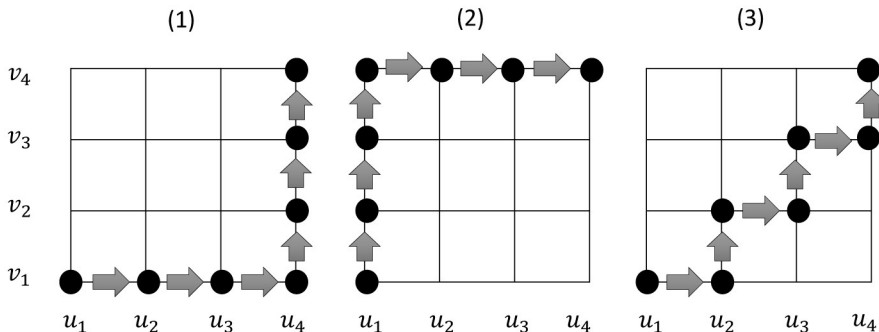

**Fig 2. Different ways to pre-select a string of drug combinations with monotonically increasing toxicity for the initial titration design.**

**Theorem 6** *Assume that the initial design and the model-based design are coherent. Let M denote the first patient whose dose is selected based on the model-based design, i.e., the patient enrolled at the moment of transition from the initial design to the model-based design. If* $(Y_M - p_T)\{F(X^*_{M+1}, \boldsymbol{\theta}) - F(X^*_M, \boldsymbol{\theta})\} \le 0$ *almost surely, then the design with* $X^*_n$ *is coherent.*

## Application

We apply our results to examine the coherence of some model-based drug combination designs in the literature.

**Example 1**. **Logistic and scaled logistic regression models** Riviere et al. [9] considered a standard logistic regression model with $F(u_j, v_k, \boldsymbol{\theta}) = \text{logit}^{-1}(\theta_1 + \theta_2 u_j + \theta_3 v_k + \theta_4 u_j v_k)$, where $\boldsymbol{\theta} = (\theta_1, \theta_2, \theta_3, \theta_4) \in \mathbb{R}^4$, with $\theta_2 > 0$ and $\theta_3 > 0$ ensuring that the toxicity probability is increasing with the increasing dose level of each agent alone, and $\theta_3 + \theta_4 u_j > 0$ for all $j$ and $\theta_2 + \theta_4 v_k > 0$ for all $k$ ensuring that the toxicity probability is increasing with the increasing dose levels of both agents together.

Depending on how the doses $u_j$ and $v_k$ are specified, Theorem 1 or Theorem 3 can be used to examine the coherence of the logistic model-based designs. When the raw dosages of the drugs are used, $u_j > 0$ and $v_k > 0$ and thus $g(u_j, v_k, \boldsymbol{\theta}) \equiv \theta_1 + \theta_2 u_j + \theta_3 v_k + \theta_4 u_j v_k$ is increasing in $\theta_t$ for $t = 1, \ldots, 4$. Because $\text{logit}^{-1}(\cdot)$ is monotonically increasing, it follows that $F(u, v, \boldsymbol{\theta})$ is monotonically increasing in $\theta_t$ for $t = 1, \ldots, 4$, and thus the uniform monotonicity condition holds. By Theorem 3, the ND-design with the logistic regression model is coherent. Also, by Theorem 5, the D-design with the logistic regression model is weakly coherent. For the D-design to be strongly coherent, we need to examine conditions B1 and B2. Condition B1 holds because of $\theta_2 > 0$ and $\theta_3 > 0$, but it is difficult to verify condition B2 because it involves the true dose-toxicity relationship that is unknown.

In many cases, standardized doses, rather than the raw dosages, are often used in the logistic model to improve the numerical stability or interpretation. For example, Riviere et al. [9] suggested using the standardized doses $u_j = \text{logit}(a_j)$ and $v_j = \text{logit}(b_k)$, where $a_j$ and $b_k$ are estimates of the toxicity probabilities of the $j$th dose level of agent 1 and the $k$th dose level of agent 2, respectively, when they are administered individually as a single agent. In addition, logarithmic transformation or centering is often employed in practice to standardize the dose. As a result, the standardized doses $u_j$'s and $v_k$'s are not necessarily all positive. Thus, the uniform monotonicity condition does not hold and Theorem 3 cannot be used. In these cases, Theorem

1 provides a more general tool to use for studying the coherence by examining condition A, described briefly as follows. Because of the symmetric role of $u_j$ and $v_k$, without loss of generality, we assume $u_j \geq 0$ and $v_k \leq 0$. Suppose that $Y_n = 0$, define $\phi^{(t)} = (\phi_1^{(t)}, \phi_2^{(t)}, \phi_3^{(t)}, \phi_4^{(t)})$ and $\psi^{(t)} = (\psi_1^{(t)}, \psi_2^{(t)}, \psi_3^{(t)}, \psi_4^{(t)})$, where $\phi^{(t)} = \psi^{(t)}$ except for the $t$th element, $t = 1, \ldots, 4$. Let $\hat{\theta}_{n,t}$ be the posterior mean of the $t$th element of $\boldsymbol{\theta}$ based on the first $n$ observations. By the approach used in the proof of Lemma 2, for $t = 1, 2$, we have $(\phi_t^{(t)} - \psi_t^{(t)})\{F(u_j, v_k, \psi^{(t)}) - F(u_j, v_k, \phi^{(t)})\}$ $\leq 0$ implying $\hat{\theta}_{n,t} \leq \hat{\theta}_{n-1,t}$, and for $t = 3, 4$, we have $(\phi_t^{(t)} - \psi_t^{(t)})\{F(u_j, v_k, \psi^{(t)}) - F(u_j, v_k, \phi^{(t)})\}$ $\geq 0$, implying that $\hat{\theta}_{n,t} \geq \hat{\theta}_{n-1,t}$. So, $(\hat{\theta}_{n,t} - \hat{\theta}_{n-1,t})\partial F/\partial \theta_t \leq 0$ for $t = 1, 2, 3, 4$. Similarly, we obtain $(\hat{\theta}_{n,t} - \hat{\theta}_{n-1,t})\partial F/\partial \theta_t \geq 0$ for $t = 1, 2, 3, 4$ when $Y_n = 1$. Therefore, condition A holds, and we obtain a more general result: the ND-design with the logistic regression model is coherent, and the D-design with the logistic regression model is weakly coherent, no matter how $u_j$ and $v_k$ are specified.

Braun and Jia [8] proposed different coding for the doses, specifying dose 1 as a categorical dummy variable and dose 2 as a continuous variable. They called the resulting model the generalized CRM model, given by

$$\text{logit}\{F_{j,k}(\boldsymbol{\theta})\} = \alpha_k + \beta u_j,$$

where $-\infty < \alpha_k < \infty$, $k = 1, \ldots, K$ and $\beta > 0$. Cai et al. [10] proposed a modification of the logistic model, namely the scaled logistic model, for drug combination trials that involve molecularly targeted agents, with the form

$$F(u_j, v_k, \boldsymbol{\theta}) = \rho \, \text{logit}^{-1}(\theta_1 + \theta_2 u_j + \theta_3 v_k),$$

where $0 < \rho < 1$. The scaled logistic model plateaus at $\rho$, rather than 1 as in the standard logistic model. It can be shown that the ND-design with the generalized CRM model or scaled logistic regression model is coherent, and the D-design with the generalized CRM or scaled logistic regression model is weakly coherent. The details are provided in the Appendix B of Supplemental material.

**Example 2**. **Change-point model** For some targeted agent, the dose-toxicity curve may initially increase at low doses and then plateau at high doses. Cai et al. [10], Riviere et al. [11] and Sato et al. [12] proposed using the change-point model for some drug combination trials, as given by

$$\text{logit}\{F_{j,k}(\boldsymbol{\theta})\} = (\alpha + \beta u_j + \gamma v_k)I(\alpha + \beta u_j + \gamma v_k \leq w) + wI(\alpha + \beta u_j + \gamma v_k > w),$$

where $I(\cdot)$ denotes an indicator function, $\boldsymbol{\theta} = (\alpha, \beta, \gamma, w)$ with $-\infty < \alpha < \infty$, $\beta > 0$, $\gamma > 0$ and $-\infty < w < \infty$. The curve of the model initially increases with the dose level but flattens once it passes the threshold defined by $\alpha + \beta u_j + \gamma v_k = w$. Let $\eta(x) = \text{logit}(x)$ for $0 < x < 1$. Suppose that $Y_n = 0$, in what follows, we use $\alpha$ as an example to show that condition A holds. Let $\hat{\alpha}_n$ denote a posterior mean of $\alpha$ based on the first $n$ patients. For fixed values of $\beta$, $\gamma$ and $w$, we take $\phi = (\alpha_1, \beta, \gamma, w)$ and $\psi = (\alpha_2, \beta, \gamma, w)$ for any $\alpha_1$ and $\alpha_2$. By the mean value theorem, we have $F_{j,k}(\phi) - F_{j,k}(\psi) = [\eta\{F_{j,k}(\phi)\} - \eta\{F_{j,k}(\psi)\}]/\eta'(\tilde{F})$, where $\tilde{F}$ lies between $F_{j,k}(\phi)$, and $F_{j,k}(\psi)$ and $\eta'(F)$ denotes the derivative of $\eta$ with respect to $F$. For $\alpha_1 \leq \alpha_2$, there are three possible cases: (1) $\alpha_1 + \beta u_j + \gamma v_k \leq w$ and $\alpha_2 + \beta u_j + \gamma v_k \leq w$; (2) $\alpha_1 + \beta u_j + \gamma v_k \leq w$ and $\alpha_2 + \beta u_j + \gamma v_k > w$; and (3) $\alpha_1 + \beta u_j + \gamma v_k > w$. In the first case, $\eta\{F_{j,k}(\phi)\} - \eta\{F_{j,k}(\psi)\} = \alpha_1 - \alpha_2 \leq 0$. The second case yields $\eta\{F_{j,k}(\phi)\} - \eta\{F_{j,k}(\psi)\} = \alpha_1 + \beta u_j + \gamma v_k - w \leq 0$, and the third case induces $\eta\{F_{j,k}(\phi)\} - \eta\{F_{j,k}(\psi)\} = w - w = 0$. That is, $F$ is nondecreasing in $\alpha$, noting that $\eta'(x) = 1/\{x(1-x)\} \geq 0$ for $0 < x < 1$. Thus, we have $(\alpha_1 - \alpha_2)\{F_{j,k}(\psi) - F_{j,k}(\phi)\} \leq 0$ for any $\alpha_1$ and $\alpha_2$. By the

approach used in the proof of Lemma 2, we have $\hat{\alpha}_n \leq \hat{\alpha}_{n-1}$ and $(\hat{\alpha}_n - \hat{\alpha}_{n-1})\partial F/\partial\alpha \leq 0$. Along a similar line, it can be shown that a similar inequality holds for $\beta$, $\gamma$ and $w$ (see the Appendix B of Supplemental material for details). Similarly, we obtain the inequality for all parameters when $Y_n = 1$, and thus condition A holds. Applying Theorem 1 and Theorem 5, we conclude that the ND-design with the change-point model is coherent and the D-design with the change-point model is weakly coherent, respectively.

**Example 3**. **Copula-type regression models** Yin and Yuan [5] proposed a drug combination design based on the Clayton copula regression model

$$F_{j,k}(\boldsymbol{\theta}) = 1 - \left\{ (1 - p_j^\alpha)^{-\gamma} + (1 - q_k^\beta)^{-\gamma} - 1 \right\}^{-1/\gamma},$$

where $p_j$ and $q_k$ are prior estimates of the toxicity probability for level $j$ of agent 1 and level $k$ of agent 2, respectively, when they are used as monotherapy, and $\boldsymbol{\theta} = (\alpha, \beta, \gamma)$ with $\alpha, \beta, \gamma > 0$. Let $G(x) = (1 - x)^{-\gamma}$. Then, $G\{F_{j,k}(\boldsymbol{\theta})\} = \{(1 - F_{j,k}(\boldsymbol{\theta}))\}^{-\gamma} = (1 - p_j^\alpha)^{-\gamma} + (1 - q_k^\beta)^{-\gamma} - 1$. Suppose that $Y_n = 0$. To apply Theorem 1, we check condition A with respect to $\alpha$. For fixed values of $\beta$ and $\gamma$, we take $\phi = (\alpha_1, \beta, \gamma)$ and $\psi = (\alpha_2, \beta, \gamma)$ for any $\alpha_1$ and $\alpha_2$. By the mean value theorem, we have $F_{j,k}(\phi) - F_{j,k}(\psi) = \{(1 - p_j^{\alpha_1})^{-\gamma} - (1 - p_j^{\alpha_2})^{-\gamma}\}/G'(\tilde{F})$ for some $\tilde{F}$ that lies between $F_{j,k}(\phi)$ and $F_{j,k}(\psi)$, where $G'(F)$ denotes the derivative of $G$ with respect to $F$. Since $G(x)$ is increasing and $f(x) = (1 - p^x)^{-\gamma}$ for some $p \in (0, 1)$ is decreasing, $F_{j,k}(\phi) - F_{j,k}(\psi) \geq 0$ for $\alpha_1 \leq \alpha_2$, i.e., $F$ is nonincreasing in $\alpha$. Let $\hat{\alpha}_n$ be a posterior mean of $\alpha$ based on the first $n$ patients. By using the approach used in the proof of Lemma 2, we have $\hat{\alpha}_n \geq \hat{\alpha}_{n-1}$ and thus $(\hat{\alpha}_n - \hat{\alpha}_{n-1})\partial F/\partial\alpha \leq 0$. We can show that a similar inequality holds with respect to $\beta$ and $\gamma$ (see the Appendix B of Supplemental material). Thus, condition A holds. Therefore, the drug combination ND-design based on the Clayton copula regression model is coherent and the D-design with the Clayton copula regression model is weakly coherent.

For drug combination trials, Yin and Yuan [5] proposed an alternative copula-type regression model, i.e., the Gumbel model, as given by $F_{j,k}(\boldsymbol{\theta}) = 1 - (1 - p_j^\alpha)(1 - q_k^\beta)$ $\{1 + p_j^\alpha q_k^\beta - 2p_j^\alpha q_k^\beta (e^\gamma + 1)^{-1}\}$ where $\boldsymbol{\theta} = (\alpha, \beta, \gamma)$ with $\alpha, \beta, \gamma > 0$. As shown in the Appendix B of Supplemental material, the ND-design based on the Gumbel model is also coherent, and the D-design based on the Gumbel model is weakly coherent.

**Example 4**. **Log-linear model** Wang and Ivanova [4] considered a toxicity model given by

$$\log\{1 - F_{j,k}(\theta)\} = \alpha \log(1 - u_j) + \beta \log(1 - v_k) + \gamma \log(1 - u_j)\log(1 - v_k)$$

with $\boldsymbol{\theta} = (\alpha, \beta, \gamma)$, where $\alpha > 0$, $\beta > 0$ and $\gamma < 0$. Let $G_{j,k}(\boldsymbol{\theta}) = \alpha \log(1 - u_j) + \beta \log(1 - v_k) + \gamma \log(1 - u_j)\log(1 - v_k)$. Then, $F_{j,k}(\boldsymbol{\theta}) = 1 - \exp\{G_{j,k}(\boldsymbol{\theta})\}$. Suppose that $Y_n = 0$. We claim that condition A holds for the design with the log-linear model with respect to $\alpha$. Let $\hat{\alpha}_n$ denote a posterior mean of $\alpha$ based on the first $n$ patients. For fixed values of $\beta$ and $\gamma$, we take $\phi = (\alpha_1, \beta, \gamma)$ and $\psi = (\alpha_2, \beta, \gamma)$ for any $\alpha_1$ and $\alpha_2$. Then, $F_{j,k}(\phi) - F_{j,k}(\psi) = \{G_{j,k}(\psi) - G_{j,k}(\phi)\}\exp(\tilde{G})$ for some $\tilde{G}$ that lies between $G_{j,k}(\phi)$ and $G_{j,k}(\psi)$. If $u_j < 0$, then $G_{j,k}(\psi) - G_{j,k}(\phi) \geq 0$ for $\alpha_1 \leq \alpha_2$, implying that $F_{j,k}(\phi) - F_{j,k}(\psi) \geq 0$ for $\alpha_1 \leq \alpha_2$, i.e., $F$ is nondecreasing in $\alpha$. By the approach used in the proof of Lemma 2, $\hat{\alpha}_n \geq \hat{\alpha}_{n-1}$. So, $(\hat{\alpha}_n - \hat{\alpha}_{n-1})\partial F/\partial\alpha \leq 0$. Likewise, $(\hat{\alpha}_n - \hat{\alpha}_{n-1})\partial F/\partial\alpha \leq 0$ is obtained even if $0 \leq u_j(< 1)$. In other words, regardless of the sign of the dose for the first agent $u_j$, we obtain $(\hat{\alpha}_n - \hat{\alpha}_{n-1})\partial F/\partial\alpha \leq 0$. Similarly, we can show that the inequality, Eq (3), holds for $\beta$ and $\gamma$ (see the Appendix B of Supplemental material for details), and thus condition A holds. So, the ND-design is coherent and the D-design is weakly coherent.

**Example 5**. **Six-parameter model** Thall et al. [3] considered a six-parameter toxicity model given by

$$F_{j,k}(\boldsymbol{\theta}) = \frac{\alpha_1 u_j^{\beta_1} + \alpha_2 v_k^{\beta_2} + \alpha_3 (u_j^{\beta_1} v_k^{\beta_2})^{\beta_3}}{1 + \alpha_1 u_j^{\beta_1} + \alpha_2 v_k^{\beta_2} + \alpha_3 (u_j^{\beta_1} v_k^{\beta_2})^{\beta_3}},$$

where $0 \leq u_j \leq 1$ and $0 \leq v_k \leq 1$ are standardized doses, and $\boldsymbol{\theta} = (\alpha_1, \alpha_2, \alpha_3, \beta_1, \beta_2, \beta_3)$ with $\alpha_1 > 0, \alpha_2 > 0, \alpha_3 (v_k^{\beta_2})^{\beta_3} > 0$, $k = 1, \ldots, K$ and $\alpha_3 (u_j^{\beta_1})^{\beta_3} > 0$, $j = 1, \ldots, J$, to ensure that toxicity monotonically increases with each of the doses. Let $G_{j,k}(\boldsymbol{\theta}) = \alpha_1 u_j^{\beta_1} + \alpha_2 v_k^{\beta_2} + \alpha_3 (u_j^{\beta_1} v_k^{\beta_2})^{\beta_3}$. Then, $F_{j,k}(\boldsymbol{\theta}) = G_{j,k}(\boldsymbol{\theta})/\{1 + G_{j,k}(\boldsymbol{\theta})\}$. Suppose that $Y_n = 0$. It is easy to see that $\partial G/\partial \alpha_1 = u_j^{\beta_1} \geq 0$ and $F$ is nondecreasing in $\alpha_1$, because $f(x) = x/(1 + x)$ is nondecreasing in $x$. Let $\hat{\alpha}_{n,1}$ denote the posterior mean of $\alpha_1$ based on the first $n$ patients. By the approach used in the proof of Lemma 2, $\hat{\alpha}_{n,1} \leq \hat{\alpha}_{n-1,1}$ and $(\hat{\alpha}_{n,1} - \hat{\alpha}_{n-1,1}) \partial F/\partial \alpha_1 \leq 0$. We can show that a similar inequality, Eq (3), holds for $\alpha_2, \alpha_3, \beta_1, \beta_2$ and $\beta_3$. Therefore, condition A holds and, under the six-parameter model, the ND-design is coherent and the D-design is weakly coherent.

## Discussion

Drug combination trials are more challenging than single-agent trials because of the higher dimensions of the dose search space and partial ordering among the drug combinations. We have proposed the concept of coherence for drug combination trials and distinguished two types of drug combination designs: the ND-design, which forbids diagonal dose movement, and the D-design, which allows for diagonal dose movement. To account for the possibility of model misspecification when using the D-design, we further defined weak coherence and strong coherence based on the model estimates and true toxicity probabilities of the doses, respectively. We provided sufficient conditions to study the coherence of the model-based drug combination designs. We investigated a number of drug combination models in the literature and showed that under these models, the ND-design is coherent and the D-design is weakly coherent. In general, it is difficult to establish strong coherence of a D-design because it involves the knowledge of the unknown, true dose-toxicity relationship. From a practical viewpoint, if strong coherence is desirable, we recommend adopting the ND-design by forbidding diagonal dose movement.

We have shown that some model-based designs are coherent for ND-designs and weakly coherent for D-designs, however not all model-based designs are coherent. One example is partial-order CRM [7], which fits multiple (single-agent) CRM models, each assuming a different toxicity order for the combinations, and then uses Bayesian model averaging (BMA) to summarize the estimates over different models and make the decision of dose transition. Although each CRM model is coherent, averaging over them would lead to incoherent decisions. This is because each of the CRM models may recommend a different dose-transition decision, and the final decision based on BMA would be consistent with some, but inconsistent with others, leading to incoherence.

We assumed that the toxicity outcomes of previous patients have been fully observed before accruing the next cohort of patients. This however may not be true when the toxicity is late-onset. In this case, we can employ the Bayesian data augmentation method to handle the delayed toxicity and facilitate real-time decision making of dose escalation and de-escalation [13, 14]. The coherence of drug combination designs in the presence of late-onset toxicity will be an interesting topic of future research.

## Supporting information

**S1 File. Appendix A and B of supplemental material are available with this paper at PLOS ONE website.**
(PDF)

## Author Contributions

**Investigation:** Yeonhee Park.

**Methodology:** Yeonhee Park.

**Writing – original draft:** Yeonhee Park.

**Writing – review & editing:** Yeonhee Park, Suyu Liu.

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
