## [Decision Letter · Decision Letter 0]

30 Sep 2020

PONE-D-20-25012

On the coherence of model-based dose-finding designs for drug combination trials

PLOS ONE

Dear Dr. Liu,

Thank you for submitting your manuscript to PLOS ONE. After careful consideration, we feel that it has merit but does not fully meet PLOS ONE’s publication criteria as it currently stands. Therefore, we invite you to submit a revised version of the manuscript that addresses the points raised during the review process.

Please attend to reviewer 2's minor comments.

We look forward to receiving your revised manuscript.

Kind regards,

Alan D Hutson

Academic Editor

PLOS ONE

Journal Requirements:

Additional Editor Comments (if provided):

Please attend to reviewer 2's minor comments.

Reviewers' comments:

Reviewer's Responses to Questions

**Comments to the Author**

1. Is the manuscript technically sound, and do the data support the conclusions?

Reviewer #1: Yes

Reviewer #2: Yes

2. Has the statistical analysis been performed appropriately and rigorously? 

Reviewer #1: N/A

Reviewer #2: Yes

3. Have the authors made all data underlying the findings in their manuscript fully available?

Reviewer #1: Yes

Reviewer #2: Yes

4. Is the manuscript presented in an intelligible fashion and written in standard English?

Reviewer #1: Yes

Reviewer #2: Yes

5. Review Comments to the Author

Reviewer #1: This paper has been well written. From my point of view , defining a coherence conditionnally to the scenario has no interest as the scenario is always unknown. However, the application of ND coherence may be useful to biostatistician working on the protocol of early phase dose finding study.

Reviewer #2: This is a very well written paper that describes the concept of coherence in drug combination trials and provides sufficient conditions for a model-based design to be coherent. As drug combinations studies are increasingly used in treating cancer, this papaer adds an important element to the literature on the coherence of dose finding designs. I have a few very minor comments.

1. The authors considered some model-based designs in the literature. Could the authors briefly comment on if there are any commonly used designs in the literature that are not coherent?

2. Could the authors briefly comment on how to deal with delayed toxicity?

6. PLOS authors have the option to publish the peer review history of their article (what does this mean?). If published, this will include your full peer review and any attached files.

Reviewer #1: No

Reviewer #2: No

---

## [Author Response · Author response to Decision Letter 0]

23 Oct 2020

Point-by-point responses to comments from the reviewers:

Reviewer #1: This paper has been well written. From my point of view, defining a coherence conditionally to the scenario has no interest as the scenario is always unknown. However, the application of ND coherence may be useful to biostatistician working on the protocol of early phase dose finding study.

Reviewer #2: This is a very well written paper that describes the concept of coherence in drug combination trials and provides sufficient conditions for a model-based design to be coherent. As drug combinations studies are increasingly used in treating cancer, this paper adds an important element to the literature on the coherence of dose finding designs. I have a few very minor comments.

1. The authors considered some model-based designs in the literature. Could the authors briefly comment on if there are any commonly used designs in the literature that are not coherent?

Response: Thank you for comment. One example is partial-order CRM (Wages et al. 2011), which fits multiple (single-agent) CRM models, each assuming a different toxicity order for the combinations, and then uses Bayesian model averaging (BMA) to summarize the estimates over different models and make the decision of dose transition. Although each CRM model is coherent, averaging over them would lead to incoherent decisions. This is because each of the CRM models may recommend a different dose-transition decision, and the final decision based on BMA would be consistent with some, but inconsistent with others, leading to incoherence. We have addressed this point in Discussion, Section 4. 

2. Could the authors briefly comment on how to deal with delayed toxicity?

Response: Thank you for comment. Bayesian data augmentation can be adopted to handle the delayed toxicity and facilitate real-time decision making of dose escalation and de-escalation (Liu and Ning, 2013 and Liu et al. 2013). We have added comments on this issue to Discussion, Section 4. 

Here are references mentioned above.

Liu S, Ning J. A Bayesian dose-finding design for drug combination trials with delayed toxicities. Bayesian analysis. 2013 Sep;8(3):703.

Liu S, Yin G, Yuan Y. Bayesian data augmentation dose finding with continual reassessment method and delayed toxicity. The annals of applied statistics. 2013 Dec 1;7(4):1837.

Wages NA, Conaway MR, O'Quigley J. Continual reassessment method for partial ordering. Biometrics. 2011 67(4):1555-63.

---

## [Decision Letter · Decision Letter 1]

5 Nov 2020

On the coherence of model-based dose-finding designs for drug combination trials

PONE-D-20-25012R1

Dear Dr. Liu,

We’re pleased to inform you that your manuscript has been judged scientifically suitable for publication and will be formally accepted for publication once it meets all outstanding technical requirements.

Kind regards,

Alan D Hutson

Academic Editor

PLOS ONE

Additional Editor Comments (optional):

Reviewers' comments:

Reviewer's Responses to Questions

**Comments to the Author**

1. If the authors have adequately addressed your comments raised in a previous round of review and you feel that this manuscript is now acceptable for publication, you may indicate that here to bypass the “Comments to the Author” section, enter your conflict of interest statement in the “Confidential to Editor” section, and submit your "Accept" recommendation.

Reviewer #2: All comments have been addressed

2. Is the manuscript technically sound, and do the data support the conclusions?

Reviewer #2: Yes

3. Has the statistical analysis been performed appropriately and rigorously? 

Reviewer #2: Yes

4. Have the authors made all data underlying the findings in their manuscript fully available?

Reviewer #2: Yes

5. Is the manuscript presented in an intelligible fashion and written in standard English?

Reviewer #2: Yes

6. Review Comments to the Author

Reviewer #2: (No Response)

7. PLOS authors have the option to publish the peer review history of their article (what does this mean?). If published, this will include your full peer review and any attached files.

Reviewer #2: No

---

## [Editor Report · Acceptance letter]

13 Nov 2020

PONE-D-20-25012R1 

On the coherence of model-based dose-finding designs for drug combination trials 

Dear Dr. Liu:

I'm pleased to inform you that your manuscript has been deemed suitable for publication in PLOS ONE. Congratulations! Your manuscript is now with our production department. 

Kind regards, 

on behalf of

Dr. Alan D Hutson 

Academic Editor

PLOS ONE